# Identification and Structure–Activity Relationship of Recovered Phenolics with Antioxidant and Antihyperglycemic Potential from Sugarcane Molasses Vinasse

**DOI:** 10.3390/foods11193131

**Published:** 2022-10-08

**Authors:** Zhe Huang, Yinning Chen, Riming Huang, Zhengang Zhao

**Affiliations:** 1School of Food Science and Engineering, South China University of Technology, Guangzhou 510640, China; 2College of Light Industry and Food Engineering, Guangxi University, Nanning 530004, China; 3Guangdong Provincial Key Laboratory of Food Quality and Safety, College of Food Science, South China Agricultural University, Guangzhou 510640, China; 4Overseas Expertise Introduction Center for Discipline Innovation of Food Nutrition and Human Health (111 Center), Guangzhou 510640, China

**Keywords:** molasses vinasse, phenolic acids, recycling, antioxidant, antihyperglycemic, structure–activity relationship

## Abstract

Sugarcane molasses vinasse is the residue of the fermentation of molasses and the water and soil environmental pollutants from distilleries. However, its recycling value has been neglected. The chemical analysis of the molasses vinasse led to the isolation of a new benzoyl chloride called 2,3,4-trihydroxy-5-methoxy benzoyl chloride, as well as thirteen known compounds, including six benzoic acids. The structure of the new benzoyl chloride was elucidated on the basis of extensive spectroscopic analysis. The antioxidant activity of all isolated compounds was measured using the ORAC assay. Moreover, we compared the cellular antioxidant activity (CAA) and inhibitory activity against α-amylase and α-glucosidase for structure–activity analysis. The results showed that only vanillic acid had CAA (8.64 μmol QE/100 μmol in the no PBS wash protocol and 6.18 μmol QE/100 μmol in the PBS wash protocol), although other benzoic acid derivatives had high ORAC values ranging between 1879.9 and 32,648.1 μmol TE/g. Additional methoxy groups at the *ortho*-positions of the *p*-hydroxy group of benzoic acids enhanced the inhibition of α-glucosidase but reduced the ORAC activity unless at the *para*-position. This work indicated that phenolics, especially phenolic acids in the sugarcane molasses vinasse, possessed potential antioxidant and antihyperglycemic activity, which improved the utilization rate of resources and reduced the discharge of pollutants.

## 1. Introduction

Molasses is a by-product of processing sugarcane in the sugar mill. Molasses has been used as an important raw material for alcohol fermentation from plants due to its huge annual production and low recovery price. Molasses vinasse is the residue of the fermentation and distillation of alcohol from sugar cane and beet molasses, which has a deep brown color; high BOD of the raw spent wash; low pH with high concentrations of organic matter such as crude proteins, lactic acid, glycerol, cholesterol, amino acid and reducing sugars [1,2,3]; and it also contains a high concentration of phenolic compounds and pigments [4]. Worldwide, 22.4 gigaliters of sugarcane molasses vinasse are produced annually, which is 14 times as much as the annual production of sugarcane-based bioethanol [5]. After molasses fermentation, the main organic compounds in the residue include sugars (glucose and fructose), amino acids (glutamic and aspartic acid), alcohols (3-methoxy-4-hydroxyphenyl glycerol and glycerol) and acids (pyroglutamic acid and *p*-hydroxybenzoic acid) [6]. The bioethanol is separated by distillation and collected through molecular sieves while glycerol and organic acids are retained in the molasses vinasse, thus, offering the possibility to recover valuable acids [7]. Phenolic acids are rich in sugarcane and its by-products, and plant-derived phenolics, including phenolic acids and anthocyanins, play key or complementary roles in the treatment of cancer, cardiovascular and chronic diseases [8].

Recently, researchers are beginning to focus on improving resource efficiency by recovering valuable active substances from agricultural or agro-processing by-products. Bio-active compounds with potential activities to scavenge free radicals, alleviate oxidative stress, regulate glucose/lipid metabolism and the immune system and fight tumors are present in abundant wastes. In China, the molasses vinasse is being abandoned by many large-scale distilleries integrated with the sugar mill, resulting in both wasted resources and pollution [9]. Nevertheless, plenty of studies focused on phenolic constituents in the sugarcane bagasse and rind, and little information on the compositions and potential bioactivity of phenolic acids from molasses vinasse is available [10]. Furthermore, high levels of dietary intake of phenolic acids are strongly correlated with activities such as relieving oxidative stress and improving diabetes in humans [11,12]. Therefore, our goal was to extract and purify the phenolic acids from molasses vinasse by column chromatography and semi-preparative HPLC and assess their antioxidant activity by ORAC and CAA assay, antihyperglycemic activity by inhibition assay of α-amylase and α-glucosidase. This research aims to provide structure information on phenolic compounds with antioxidant and anti-diabetic activity in molasses vinasse and can be considered as suggestions and references for large-scale recovery.

## 2. Materials and Methods

### 2.1. Materials and Reagents

Molasses vinasse was obtained from Changling Sugar Mill (Fangchenggang, China) in November 2016. Organic solvents used for extraction were all analytical grade reagents.

### 2.2. Preparation of Extracts

The molasses vinasse (100 kg) was leached three times with ethyl acetate (1:1) at room temperature. The ethyl acetate solutions were combined and concentrated to afford chartreuse oil (53.6 g).

### 2.3. Extraction and Isolation

The n-hexane, acetone and MeOH (20:4:3) eluent were used in a column chromatograph on a column of 200–300 mesh silica gel to separate the EtOAc-soluble oil into seven fractions (PE-1–PE-7). Fraction PE-1 was chromatographed on a silica gel column and eluted with n-hexane: acetone (5:1) to yield six sub-fractions (PE-1-1~PE-1-6). There was a transparent crystal that appeared in fraction PE-2, then the crystal was filtered with n-hexane and recrystallized twice with chloroform to get compound **6** (25.8 mg). Fraction PE-7 was subjected to further chromatography on a silica gel column with CHCl_3_: MeOH eluting system 10:1 to yield seven sub-fractions (PE-7-1~PE-7-7). Fraction 7-1 was subfractionated with column chromatography gel on Sephadex LH-20 gel (75–150 μm), using CHCl_3_:MeOH (10:1) to yield two subfractions (PE-7-1-1 and PE-7-1-2). Compound **2** (10.3 mg) was isolated by further separating fraction PE-7-1-1 using Sephadex LH-20 and MeOH. Fraction PE-7-3 was subjected to silica gel CC eluted with a gradient of n-hexane/acetone (5:1) to furnish two sub-fractions (PE-7-3-1 and PE-7-3-2). Fraction PE-7-3-2 was found as a massive transparent crystal and was purified by recrystallization with MeOH to yield **3** (27.3 mg). Fraction PE-1-6 was separated by RP-HPLC, using the mixtures of MeOH and H_2_O (MeOH:H_2_O = 19:81) to yield **4** (3.1 mg), **5** (11.5 mg), **1** (12.0 mg) and **7** (14.3 mg). Then, **13** (38.5 mg) was obtained from fraction PE-7-1-2 by using semi-preparative HPLC (Waters 600E liquid chromatograph, Waters Corp., Milford, MA, USA), with MeOH:H_2_O (40:60) as eluent. Fraction PE-7-2 was separated by RP-HPLC using an isocratic elution of MeOH and H_2_O (MeOH:H_2_O = 10:90) to yield **14** (2.5 mg). Fraction PE-7-7 was further purified by RP-HPLC eluted in MeOH and H_2_O (MeOH:H_2_O = 18:82) to generate **9** (8.6 mg), **10** (2.5 mg), **11** (14.3 mg) and **12** (6.0 mg).

### 2.4. Oxygen Radical Scavenging Capacity (ORAC) Assay

The antioxidant activity of compounds was measured using the ORAC assay according to a previous method [13]. Briefly, 150 μL 8.16 × 10^−5^ mM fluorescein working solution was added to the inner wells of a 96-well microplate. Then the plate was incubated at 37 °C for 10 min, after which another 25 μL diluted antioxidants were added using Trolox as standard and 75 mM buffer solution as blank. The plate was shaken every 3 min during the incubation. Then, followed by the addition of 25 μL of 153 mM AAPH to generate the peroxyl radicals, the fluorescence was recorded on a Victor3 96-well plate reader (PerkinElmer Inc., Norwalk, CT, USA) for 1 min for each of 35 cycles. All samples were analyzed in triplicate at three different dilutions. ORAC values were expressed as mean micromoles of Trolox equivalents (TE) per 1 g of the compound.

### 2.5. Cellular Antioxidant Activity (CAA) Assay

HepG2 cells (ATCC Co., Manassas, VA, USA) were incubated in DMEM with 1% penicillin-streptomycin and 10% FBS and maintained at 37 °C with 5% CO_2_. The modified methylene blue assay was used to determine cytotoxicity [14].

The CAA assay was performed, according to Wolfe et al. [15], with slight modification. In brief, 6 × 10^4^ HepG2 cells were placed in each well of a black-walled 96-microwell plate. 200 μL of PBS was added to the edge of the plate to prevent the edge effect. After adherence, cells were treated with a range of concentrations of benzoic acid derivatives and a standard (quercetin) containing 50 μM DCFH-DA for 1 h under the same culture condition. After removing the supernatant medium, the cells were washed with or without PBS. Then promptly added 600 μM ABAP into each well except blank groups before reading at 485 nm excitation and 538 nm emission using a multifunctional microplate detector (Molecular Devices, Sunnyvale, CA, USA). The fluorescence values were recorded every 5 min for 1 h. The CAA values were obtained by converting the EC_50_ values and presented as μmol quercetin equivalents (QE)/100 μmol.

### 2.6. Enzyme Inhibition Assay In Vitro

#### 2.6.1. α-Amylase Inhibition Assay

The inhibition assay with the α-amylase was performed as a previously reported protocol with a slight modification [16]. Briefly, benzoic acid derivates (100 mM) and acarbose (positive control) were dissolved first in 30% (*v*/*v*) ethanol-potassium phosphate buffer (PBS). Then, 1 U mL^−1^ α-amylase (EC3.2.1.1), soluble starch solution (1%, *w*/*v*) and 3,5-dinitrosalicylic acid (DNS) were all dissolved in PBS (50 mM, pH 6.9). The mixed system contained a 500 μL sample or blank solution and 500 μL α-amylase incubated at 37 °C for 10 min, followed by the addition of 500 μL of preheated starch solution into the mixed incubated system to initiate the enzymatic hydrolysis reaction (37 °C, 10 min). Finally, 1 mL of DNS was added to halt the reaction, and the mixture was then allowed to sit in boiling water for 5 min. The absorbance was measured at 540 nm using a microplate reader (SpectraMax 190, Molecular Devices Co., Sunnyvale, CA, USA) at room temperature. The inhibition rate of the samples was calculated using the following equation:$$\mathrm{Inhibition}\;\mathrm{rate}\;(\%)=\left[1-\frac{(A_{\mathrm{sample}}-A_{\mathrm{sample}\;\mathrm{blank}})}{(A_{\mathrm{control}}-A_{\mathrm{control}\;\mathrm{blank}})}\right]\times 100$$
where *A*_sample_, *A*_sample blank_, *A*_control_, *A*_control blank_ represent the absorbance of the presence and absence of enzymes in sample solutions, and the absorbance of the presence and absence of enzymes in buffer solutions. The half-maximal inhibitory concentration (IC_50_) of each compound was obtained by regression analysis.

#### 2.6.2. α-Glucosidase Inhibition Assay

The inhibition assay with the α-glucosidase was carried out as a previously described protocol [17]. Briefly, the 1 U mL^−1^ α-glucosidase (EC3.2.1.20) and 3 mM *p*-nitrophenyl-α-D-glucopyranoside (*p*NP-α-G) were dissolved in PBS, except samples and acarbose were prepared using the method described as α-amylase inhibition assay. The incubated system contained 50 μL of samples and 100 μL of the α-glucosidase solution was added to a 96-well plate at 37 °C for 10 min. Then, 50 μL of the substrate *p*NP-α-G was added to start the reaction at 37 °C. After 10 min, 100 μL of 1 M Na_2_CO_3_ solution was added to inactivate α-glucosidase and halt the generation of substrates, and the absorbance was read at 405 nm with the micro-plate reader (SpectraMax 190, Molecular Devices Co., Sunnyvale, CA, USA). The percentage of inhibition and the values of IC_50_ of samples were calculated by the same equation as α-amylase inhibition assay.

### 2.7. Statistical Analysis

All the experiments were carried out in triplicates, and data of IC_50_ of samples were expressed as mean ± standard deviation (mean ± SD). The IC_50_ was calculated using nonlinear regression of SPSS 26.0 software (SPSS Inc., Chicago, IL, USA). A Duncan’s multiple-range test (SPSS 26.0) was used to examine the statistical difference between groups (*p* < 0.05). All figures were performed by Origin 2018 software (OriginLab, Northampton, MA, USA).

## 3. Results and Discussion

### 3.1. Structural Elucidation of a New Compound (***1***)

The HR-ESI-MS and NMR spectra of all compounds were recorded by a Bruker Bio TOF IIIQ mass spectrometer (Bruker Daltonics, Billerica, MA, USA) and a Bruker AM-600 spectrometer (Bruker BioSpin AG, Fallanden, Switzerland), respectively. Tetramethylsilane (TMS) was used as an internal reference for NMR analysis.

Compound (**1**) as shown in Figure 1A, a kind of amorphous solid, was established as C_8_H_7_O_5_Cl based on the HR-ESI-MS ([M + Na]^+^ at *m*/*z* 241.5804, calcd for C_8_H_7_O_5_NaCl, 241.5810). One methoxyl, one sp^2^ methine and six sp^2^ quaternary carbons were observed in the ^1^H, ^13^C NMR and HMQC spectra of compound **1**. The ^1^H NMR data suggested the presence of one methoxy signal at *δ*_H_ 3.87 (3 H, s, OMe-5) and one methine signal at *δ*_H_ 7.32 (1 H, s, H-6), and the ^13^C NMR spectrum showed typical signals for benzoyl moiety at *δ*_C_: 145.0 (C-5), 138.7 (C-4), 136.5 (C-2), 126.3 (C-3), 119.3 (C-1), 105.5 (C-6). In addition, the signals at *δ*_C_ 167.1 and 53.9 illustrated the presence of an acyl chloride and a methoxyl. The HMBC correlation data of methoxy protons with C-5 confirmed the position of the methoxy group at C-5. The acyl chloride group at C-1 was confirmed by the HMBC correlation of H-6 with COCl-1. Furthermore, the gross structure of compound **1** was established by HMBC correlation and comparison with those reported NMR data. Based on the above spectroscopic evidence, we proposed that **1** has the structure shown in Figure 1A and is named as 2,3,4-trihydroxy-5-methoxy benzoyl chloride, belonging to the phenolic compounds.

### 3.2. Physical Properties and Spectral Data of 13 Known Compounds

Structural elucidation of the known compounds was carried out by NMR analysis and compared with their spectroscopic data reported in the literature. The first stage is the identification of the phenolic acids present in the molasses vinasse of the research. Figure 1B shows the structures of 13 known compounds isolated from molasses vinasse, including 6 benzoic acid derivates.

Compound (**2**). White crystal, dissolved in MeOH. The ESI-MS spectrum of compound **2** exhibited an ion peak at *m*/*z* 169 [M + H]^+^ ([C_8_H_8_O_4_ + H]^+^). ^1^H NMR (CD_3_OD, 600 MHz) *δ*: 7.57 (1 H, s, H-6), 7.56 (1 H, d, *J* = 9.6 Hz, H-4), 6.84 (1 H, d, *J* = 9.6 Hz, H-3), 3.90 (3 H, s, OCH_3_-5). ^13^C NMR (150 MHz, CD_3_OD) *δ*: 168.7 (COOH), 151.3, 147.3, 123.9, 121.7, 113.9, 112.4, 55.0 (OCH_3_). On the basis of the above NMR data, compound **2** was identified as 2-hydroxy-5-methoxybenzoic acid which was consistent with literature reports [18].

Compound (**3**). White crystal, dissolved in MeOH and acetone. ^1^H NMR (CD_3_OD, 600 MHz) *δ*: 2.56 (6 H, s, 2 CH_3_). ^13^C NMR (150 MHz, CD_3_OD) *δ*: 174.8 (2 C=O), 28.4 (2 CH_3_). Due to the consistency of literature reports [19], compound **3** was identified as acetyl sulfide.

Compound (**4**). White powder, dissolved in MeOH. A molecular ion peak (*m*/*z* = 155, [M + H]^+^, [C_7_H_6_O_4_ + H]^+^) was observed in the ESI-MS spectrum. ^1^H NMR (CD_3_OD, 600 MHz) *δ*: 7.41 (1 H, d, *J* = 1.8 Hz, H-6), 7.35 (1 H, dd, *J* = 8.1, 1.8 Hz, H-4), 6.71 (1 H, d, *J* = 8.1 Hz, H-3). ^13^C NMR (CD_3_OD, 150 MHz) *δ*: 175.4 (COOH), 148.4, 145.3, 130.7, 122.9, 117.8, 115.2. Compared to the above data with the reported literature [20], compound **4** was identified as gentisic acid.

Compound (**5**). Amorphous solid, dissolved in MeOH. The ESI-MS spectrum gave a pseudo-molecular ion peak at *m*/*z* 169 ([C_8_H_8_O_4_ + H]^+^). ESI-MS *m*/*z* 169 [M + H]^+^. ^1^H NMR (CD_3_OD, 600 MHz) *δ*: 7.58 (1 H, d, *J* = 2.4 Hz, H-6), 7.49 (1 H, dd, *J* = 6.0, 1.2 Hz, H-2), 6.76 (1 H, d, *J* = 8.4 Hz, H-5), 3.88 (3 H, s, OMe-4). ^13^C NMR (CD_3_OD, 150 MHz) *δ*: 165.6 (COOH), 150.1, 148.1, 118.2, 115.2, 113.5, 56.3 (OCH_3_). Based on the above NMR data, compound **5** was identified as 3-hydroxy-4-methoxybenzoic acid (isovanillic acid) which was consistent with literature reports [21].

Compound (**6**). Acicular crystal, dissolved in CHCl_3_. ^13^C NMR (150 MHz, CD_3_OD) *δ*: 140.8, 121.6, 71.5, 56.8, 55.9, 50.2, 45.8, 42.0, 39.7, 37.2, 36.5, 36.1, 33.9, 31.9, 31.8, 31.3, 29.2, 28.9, 28.2, 25.3, 24.3, 23.0, 21.1, 19.7, 19.3, 18.9, 18.7, 12.1, 11.9. Thus, compound **6** was identified as β-sitosterol which was consistent with literature reports [22].

Compound (**7**). Amorphous solid, dissolved in MeOH. An obvious ion peak was found in the ESI-MS spectrum which indicated the molecular formula C_7_H_6_O_3_ for compound **7**. ^1^H NMR (CD_3_OD, 600 MHz) *δ*: 7.82 (2 H, d, *J* = 9.8 Hz, H-2, 6), 6.73 (2 H, d, *J* = 9.8 Hz, H-3, 5). ^13^C NMR (CD_3_OD, 150 MHz) *δ*: 169.3 (COOH), 161.1, 132.4 (C-2, 6), 129., 115.3. Due to the consistency of literature reports [23], compound **7** was identified as 4-hydroxybenzoic acid.

Compound (**8**). Amorphous solid, dissolved in MeOH. ESI-MS *m*/*z* 169 [M + H]^+^, calcd for C_8_H_8_O_4_. ^1^H NMR (CD_3_OD, 600 MHz) *δ*: 7.57 (1 H, d, *J* = 1.8 Hz, H-2), 7.51 (1 H, dd, *J* = 9, 1.8 Hz, H-6), 6.79 (1 H, d, *J* = 9 Hz, H-5), 3.88 (3 H, s, OMe-3). ^13^C NMR (CD_3_OD, 150 MHz) *δ*: 172.1, 149.6, 146.9, 126.4, 123.2, 114.0, 112.6, 54.9 (OCH_3_). Compound **8** was identified as vanillic acid which was based on the consistency of ref. [24].

Compound (**9**). White powder, dissolved in MeOH. ^1^H NMR (CD_3_OD, 600 MHz) *δ*: 8.43 (1 H, s, H-3), 8.34 (1 H, s, H-5), 3.94 (1 H, m, H-2′), 3.77 (1 H, m, H-4′a), 3.62 (1 H, m, H-4′b), 3.54 (1 H, m, H-3′), 3.17 (1 H, dd, *J* = 14.4, 3.0 Hz, H-1′), 2.87 (1 H, dd, *J* = 14.4, 9.6 Hz, H-1′), 2.54 (3 H, s, Me-2). ^13^C NMR (CD_3_OD, 150 MHz) *δ*: 154.7, 153.3, 142.0, 141.4, 74.8, 71.6, 63.2, 38.2, 19.9 (CH_3_). Compared the above data with the reported literature [25], compound **9** was identified as 2-methyl-6-(2′,3′,4′-trihydroxybutyl)-pyrazine.

Compound (**10**). White powder, dissolved in MeOH. ^1^H NMR (CD_3_OD, 600 MHz) *δ*: 6.34 (1 H, d, *J* = 2.4 Hz, H-6), 6.28 (1 H, d, *J* = 3.0 Hz, H-2), 4.77 (1 H, d, *J* = 7.8 Hz, H-1′), 3.90 (1 H, dd, *J* = 6.0, 2.4 Hz, H-6′a), 3.80 (3 H, s, OMe-5), 3.72 (3 H, s, OMe-4), 3.69 (1 H, dd, *J* = 6.0, 6.0 Hz, H-6′b), 3.46~3.34 (4 H, m, H-2′, 3′, 4′, 5′). ^13^C NMR (CD_3_OD, 150 MHz) *δ*: 155.9, 154.9, 151.9, 133.2, 102.9, 98.7, 94.8, 78.2, 78.0, 74.9, 71.5, 62.6, 61.1 (OCH_3_), 56.3 (OCH_3_). Due to the consistency of literature reports [26], compound **10** was identified as 3-hydroxy-4,5-dimethoxyphenyl-β-D-glucopyranoside.

Compound (**11**). White powder, dissolved in MeOH. ^1^H NMR (CD_3_OD, 600 MHz) *δ*: 6.85 (1 H, d, *J* = 8.4 Hz, H-5), 6.83 (1 H, d, *J* = 3.0 Hz, H-2), 6.67 (1 H, dd, *J* = 10.4, 2.4 Hz, H-6), 4.78 (1 H, d, *J* = 7.2 Hz, H-1′), 3.90 (1 H, dd, *J* = 13.8, 1.8 Hz, H-6′a), 3.81 (3 H, s, OMe-3), 3.78 (3 H, s, OMe-4), 3.69 (1 H, dd, *J* = 18.0, 6.0 Hz, H-6′b), 3.48~3.34 (4 H, m, H-2′, 3′, 4′, 5′). ^13^C NMR (CD_3_OD, 150 MHz) *δ*: 154.0, 151.1, 145.7, 113.9, 109.3, 104.0, 103.4, 78.1 (C-3′, 5′), 75.0, 71.5, 62.6, 57.1 (OCH_3_), 56.4 (OCH_3_). Compound **11** was identified as 3,4-dimethoxyphenyl-O-β-D-glucopyranoside which was based on the consistency of literature reports [27].

Compound (**12**). White Amorphous solid, dissolved in MeOH. ^1^H NMR (CD_3_OD, 600 MHz) *δ*: 6.48 (2 H, s, H-3, 5), 4.80 (1 H, d, *J* = 7.2 Hz, H-1′), 3.91 (1 H, dd, *J* = 9.6, 2.4 Hz, H-2′), 3.87 (1 H, dd, *J* = 6.0, 2.4 Hz, H-6′a), 3.81 (6 H, s, OMe-2, 6), 3.70 (3 H, s, OMe-4), 3.65 (1 H, dd, *J* = 6.0, 6.0 Hz, H-6′b), 3.45 (2 H, t, *J* = 6.6, 3.6 Hz, H-3′, 5′). ^13^C NMR (CD_3_OD, 150 MHz) *δ*: 154.6, 153.4, 129.8, 101.8, 94.7, 77.1, 76.7, 73.5, 70.3, 61.3, 59.8 (OCH_3_), 55.2 (2 OCH_3_). Due to the consistency of literature reports [28], compound **12** was identified as 2,4,6-trimethoxyphenyl-1-O-β-D-glucopyranoside.

Compound (**13**). White crystal, dissolved in MeOH. The molecular formula C_9_H_10_O_5_ of compound **13** was revealed by analysis of HR-ESI-MS (*m*/*z* 199 [M + H]^+^). ESI-MS *m*/*z* 199 [M + H]^+^. ^1^H NMR (CD_3_OD, 600 MHz) *δ*: 7.33 (2 H, s, H-2, 6), 3.88 (6 H, s, OMe-3, 5). ^13^C NMR (CD_3_OD, 150 MHz) *δ*: 173.6 (COOH), 148.5, 139.8 (C-3, 5), 127.0, 108.0 (C-2, 6), 56.6 (2 OCH_3_). Compared with the data reported in literature [29], compound **13** was identified as syringic acid.

Compound (**14**). White Amorphous solid, dissolved in MeOH. ^1^H NMR (CD_3_OD, 600 MHz) *δ*: 6.87 (4 H, s, H-2, 3, 5, 6), 4.52 (6 H, s, OMe-1, 4). ^13^C NMR (CD_3_OD, 150 MHz) *δ*: 157.3 (C-1, 4), 115.4 (C-2, 3, 5, 6), 57.1 (2 OCH_3_). Compound **14** was identified as 1,4-dimethoxybenzene which was based on the consistency of literature reports [30].

### 3.3. Antioxidant Activity of Benzoic Acid Derivates

#### 3.3.1. Chemical Antioxidant Activity

Figure 2 illustrates the ORAC values of isolated compounds **1**–**14**. The ORAC assay, as a method to quantify the antioxidant capacity of compounds or extracts, has been widely used in biological activity evaluation. The higher ORAC value of a compound reveals the potential antioxidant activity.

The values of compounds **1**–**14** are 4094.6 ± 32.1 μmol TE/g, 32648.1 ± 2181.8 μmol TE/g, 1.3 ± 0.2 μmol TE/g, 23113.1 ± 1057.9 μmol TE/g, 20705.9 ± 835.9 μmol TE/g, 21.5 ± 0.9 μmol TE/g, 14874.9 ± 599.2 μmol TE/g, 12945.8 ± 1419.3 μmol TE/g and 78.0 ± 2.9 μmol TE/g, 4177.5 ± 324.2 μmol TE/g, 56.5 ± 2.8 μmol TE/g, 80.2 ± 2.3 μmol TE/g, 1879.9 ± 44.7 μmol TE/g and 2154.2 ± 183.1 μmol TE/g, respectively. Remarkably, benzoic acid derivatives exhibited excellent ORAC activity, and 2-hydroxy-5-methoxybenzoic acid had the highest ORAC value. Recent studies have shown that the content of hydroxybenzoic acids (vanillic, syringic and gentisic acid) was positively correlated with the ORAC activity of the extracts, which indicated that the benzoic acid derivatives in this study possessed antiradical activity at the chemical level [31,32].

#### 3.3.2. Cellular Antioxidant Activity

According to the results of the ORAC assay, we found that compounds with benzoic acid structure have stronger antioxidant activity at the chemical level. Benzoic acids as phenolic acids are common in plants and possess antioxidant and antibacterial activity [33]. Therefore, we selected compounds **2**, **4**, **5**, **7**, **8** and **13** for the intracellular antioxidant assay.

Cellular Antioxidant Activity (CAA) utilizes the surface and the inside of HepG2 cells as a reaction environment, and antioxidants prevent free radicals provided by ABAP from attacking intracellular DCFH and avoiding the production of fluorescent DCF [15]. This method established the evaluation of antioxidant capacity at the cellular level, which directly reflected the protective effect of antioxidants from free radical damage under appropriate physiological conditions with consideration of cell uptake, distribution and bioavailability.

The cellular antioxidant capacity of benzoic acid derivatives was measured at non-cytotoxic concentrations in the no PBS wash and the PBS wash protocols. We noticed that when HepG2 cells were treated with a rising concentration of benzoic acid derivatives (31.25 μM to 1000 μM), only vanillic acid (**8**) gradually inhibited the fluorescence intensity generated by DCF (Figure 3A–D). After calculation, the EC_50_ of vanillic acid were 59.08 ± 3.74 μM and 209.68 ± 3.22 μM in no PBS wash and PBS wash protocols, respectively. And the CAA values of vanillic acid were further obtained, which were 8.64 ± 0.53 μmol QE/100 μmol (without PBS wash) and 6.18 ± 0.09 μmol QE/100 μmol (with PBS wash). These results indicated the antioxidant activity of vanillic acid in a cell system; meanwhile, our work confirmed the correlation analysis of a previous study, which suggested that vanillic acid in bound extracts was the main contributor to CAA of *Ehretia macrophyla* Wall [34]. Moreover, the cellular uptake rate (the CAA value of the PBS wash/no PBS wash × 100%) of vanillic acid was 71.53%, which showed that vanillic acid was well absorbed by cells and broke the intracellular peroxyl radical chain reactions caused by ABAP [35]. 

#### 3.3.3. Structure–Activity Relationship Analysis in Antioxidant Activity

The major objects of structure–activity analysis in antioxidant activity were benzoic acids and methoxylated derivatives (**2**, **4**, **5**, **7**, **8** and **13**). The ORAC value of 2-hydroxy-5-methoxybenzoic acid (**2**) was higher than vanillic acid (**8**), which referred that the *o*-hydroxyl group of benzoic acid as beneficial to enhance antioxidant capacity. The presence of an *ortho*-hydroxyl group tended to form a quinone that was easily oxidized by free radicals, thus, increasing its antioxidant capacity [36]. Based on a previous study and the ORAC values of **2** and **4** in our work, we inferred that the methoxy substitution at C-5 increased the electron cloud density of the benzene ring and reduced the ionization potential (IP) and the bond dissociation energy (BDE) of the molecule. This effect was stronger than that of the hydroxyl group introduced at the same position [37]. The lower IP and BDE values of compounds, the higher the antioxidant activity [38].

Compared with the ORAC values of 4-hydroxybenzoic acid (**7**), vanillic acid (**8**) and syringic acid (**13**), however, we observed that the antioxidant activity of *para*-hydroxybenzoic acid decreased 1.15 folds and 7.91 folds for each additional methoxy group in the *ortho*-positions of its -OH substitution. Nevertheless, it was disputable that the effect of the introduction of methoxy groups on antioxidant activity in the benzoic acids was opposite in the ORAC assay and other chemical free radical scavenging assays like the 1,1-Diphenyl-2-picrylhydrazyl (DPPH) assay, which was caused by the difference in the reaction mechanism. Sequential proton-loss electron transfer (SPLET) and single-electron transfer followed by proton transfer (SET-PT) mechanisms dominate the free radical reaction process in DPPH instead of the hydrogen atom transfer (HAT) mechanism [38]. Furthermore, SPLET and SET-PT both involve the transfer of electrons and protons between the radical and the antioxidant [39]. In the DPPH assay, methoxy substitutions decrease the proton affinity (PA) and electron transfer enthalpy (ETE) to enhance the ability of electron-donating and free radical scavenging capacity of the molecule [38], which is consistent with the reaction mechanism of SET-PT and SPLET. However, the type of ORAC is typical HAT-based assay [40]. The methoxy groups form intramolecular hydrogen bonds (IMHBs) with the adjacent hydroxyl group and increase the BDE value of the *p*-OH of benzoic acids, which directly results in higher energy dissociation of H atoms involved in forming the O-H bond [37]. Thus, OCH_3_ becomes an obstacle to the abstraction of hydrogen to a free radical.

It is noteworthy that other benzoic acid derivates showed no CAA activity, even increasing the fluorescence intensity, which was inconsistent with the above ORAC results (Figure 3E–N). This phenomenon indicated that there was no direct correlation between the in vitro and intracellular antioxidant activity of compounds. However, cellular biological systems predicted the antioxidant activity of compounds more directly than chemical systems. OCH_3_ of the flavonoids played a role in reducing cellular antioxidant activity [41], but this statement could not support our result because of the comparison of vanillic acid and p-hydroxybenzoic acid. Recently, the behavior of compounds on the biomembrane was seen as an essential factor that can be absorbed and used by biological systems [42]. Moreover, a previous study revealed that vanillic acid and *p*-hydroxybenzoic acid were more easily absorbed by an analog biomembrane (a lipid bilayer) because of their higher lipophilicity than syringic acid [43]. We noticed that benzoic acids with or without a methoxyl group in the *ortho* position of the *p*-OH showed a completely opposite trend in the fluorescence kinetics of CAA. Therefore, the specific chemical structure of vanillic acid leads to intrinsic lipophilicity and cellular antioxidant activity.

### 3.4. Antihyperglycemic Activity of Benzoic Acid Derivates

#### 3.4.1. Inhibitory Activity of α-Amylase and α-Glucosidase

α-Amylase and α-glucosidase enzymes are key hydrolytic enzymes for glucose uptake in a carbohydrate diet. α-Amylase is secreted by the salivary glands and pancreas to hydrolyze starch into oligosaccharides and dextrin, and α-glucosidase is secreted from the small intestine to further hydrolyze the oligosaccharides to glucose [44]. The moderate inhibition effect on α-amylase and α-glucosidase is an effective and reasonable pathway to manage and control postprandial blood glucose in type 2 diabetes mellitus. Phenolic compounds have been reported to possess desirable inhibitory ability against α-amylase and α-glucosidase [45], and benzoic acids and their derivatives were identified in various plant extracts [46]. Therefore, the hypoglycemic activity of benzoic acid and its derivatives could be evaluated by enzyme inhibition assay, and then the structure–activity relationship was analyzed to discover the vital groups of the benzoic acids for their bioactivities.

Due to the low water solubility of methoxy benzoic acids, the 30% (*v*/*v*) ethanol-water solvent system was employed to dissolve the benzoic acid derivatives. The ethanol volume ratio ranged from 10 to 15% when incubated, which was considered low toxicity and a weak-inhibitory effect on enzyme [47]. However, high-concentration isovanillic acid cannot be dissolved in 30% (*v*/*v*) ethanol-water system, and it was hard to evaluate its inhibitory activity of α-amylase and α-glucosidase.

As shown in As shown in Table 1, all benzoic acid derivates showed inhibitory activity on α-glucosidase, and the activity of benzoic acid derivates was in the order of: syringic acid > 2-hydroxy-5-methoxybenzoic acid > vanillic acid ≈ gentisic acid > 4-hydroxybenzoic acid, suggested that syringic acid had the most potent α-glucosidase inhibitory activity. Interestingly, only 2-hydroxy-5-methoxybenzoic acid and gentisic acid exhibited the α-amylase inhibition effect. This result was consistent with Lei Guan et al. The addition of 2-OH enhanced the α-amylase inhibition ability of benzoic acid derivatives, which may be attributed to the formation of IMHBs and the improvement of hydrophobicity [48]. Compared with the positive control, the enzyme inhibitory activity of benzoic acid derivates was significantly lower than acarbose (*p* < 0.05), which confirmed that five tested compounds possessed moderate inhibitory activity on α-glucosidase and negligible inhibition on α-amylase.

#### 3.4.2. Structure–Activity Relationship Analysis in Antihyperglycemic Activity

All tested compounds involved in the enzyme inhibition assay had the same parent structure of benzoic acid but were distinguished by the number and position of the phenolic hydroxyl and methoxy groups on the benzene ring.

The results of the inhibition assay on α-amylase suggested that benzoic acid derivates with ortho- and meta-phenolic hydroxyl groups (**2** and **4**) had the inhibitory activity on α-amylase; conversely, the presence of -OH group in para-position of benzoic acid (**7**) did not show any inhibition effect. It can be considered that the formation of IMHBs involving the o-hydroxyl group and 1-COOH is the crucial structure of the benzoic acid derivative as an α-amylase inhibitor, which improves the stability and enhances the local hydrophobicity of the molecule, thereby binding to the cavity of the hydrophobic active pocket of α-amylase [48]. In addition, the α-amylase inhibition of compounds became weaker since methoxy groups replaced -OH groups at C-5 on the benzene ring. On the other side, it could be inferred that *o*-phenolic hydroxybenzoic acid possessed stronger inhibitory activity against α-glucosidase than that with *p*-hydroxyl group according to the IC_50_ of compounds **2** and **8**. Secondly, the increasing number of the OCH_3_ groups at C-3 and C-5 on the ring contributed to the enhancement of the inhibitory activity of benzoic acids.

It could be concluded that the *o*-hydroxyl group, instead of the *p*-hydroxyl group, increased the α-amylase and α-glucosidase inhibition of benzoic acid derivatives. IMHBs formed by the COOH and the OH at positions C-1 and C-2 arouse our thinking [49]. Moreover, the formation of IMHBs locks the COOH and the OH on the same plane, which increases the molecule’s rigidity and improves its effectiveness in enzyme inhibition [50,51].

Interestingly, the role of the methoxy substituents in α-amylase inhibitory activity was contrary to that in α-glucosidase inhibition. Generally, the presence of the methoxy substituents of the flavones and chalcones could reduce the α-glucosidase inhibition activity [52,53], which was considered as the lack of hydrogen force provided by hydroxyl when compounds bound with enzymes [54]. However, the methoxyls of benzoic acids caused the improvement of the α-glucosidase inhibitory activity, which was related to the co-effect with the adjacent hydroxy groups on the ring [55]. Malunga et al. [56] obtained consistent results, and they suggested that the lone-pair electrons provided by the methoxy groups competed with enzymic substrates and occupied the enzyme’s active site. Many authors believed that while methoxy groups were introduced into benzene rings, the enhancement of the activity of compounds was attributed to a strong electron-donating effect to form the bonding affinity with enzymes [57]. However, these hypotheses did not seem to fully explain the different roles of OCH_3_ in two digestive enzymes. We referred that it was related to the difference of active sites when benzoic acid bonds with α-amylase or α-glucosidase, which required more in-depth mechanism analysis like molecular docking to prove. Our research provided the reference for screening and synthesis of α-amylase and α-glucosidase inhibitors with effective groups from benzoic acid derivates.

## 4. Conclusions

A new chloride, named 2,3,4-trihydroxy-5-methoxy benzoyl chloride (**1**), along with thirteen known compounds, including six benzoic acids, a thioether, a sterol, a pyrazine, three glucopyranosides and an anisole, were isolated from the ethyl acetate fraction of sugarcane molasses vinasse. It can be concluded that benzoic acid derivates (**2**, **4**, **5**, **7**, **8** and **13**) exhibited excellent antioxidant activity in the ORAC assay. However, contradictory results appeared in the CAA assay that only vanillic acid showed the intracellular antioxidant capability, which resulted from its appropriate lipophilicity to enter the cells. In enzyme inhibitory assays, only 2-hydroxy-5-methoxybenzoic acid and gentisic acid exhibited moderate inhibition of α-amylase and α-glucosidase. Moreover, hydroxy and methoxy groups were critical to the activity of benzoic acids: (1) *o*-OH enhanced the ORAC activity and improved the inhibition effect on α-amylase and α-glucosidase; (2) OCH_3_ at *para*-position and at *ortho*-position of OH would increase and decrease the value of ORAC, respectively; (3) the presence of OCH_3_ weakened the α-amylase inhibition activity but strengthened the α-glucosidase inhibition activity. However, the bioactivities of untested benzoic acid derivatives with different groups and positions still need to be evaluated to obtain more effective structure–activity relationships. In summary, this study provided the guiding recommendations for the recovery of bioactive ingredients from the cane sugar molasses vinasse.

## Figures and Tables

**Figure 1 foods-11-03131-f001:**
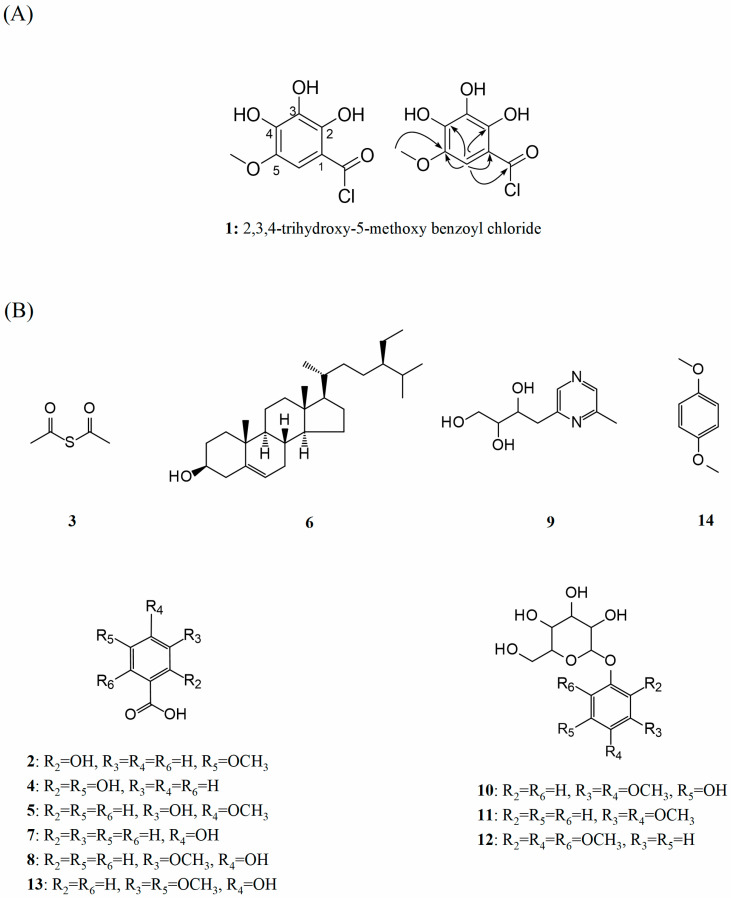
(**A**) Structure and HMBC of compound **1**; (**B**) the structures of 13 known compounds.

**Figure 2 foods-11-03131-f002:**
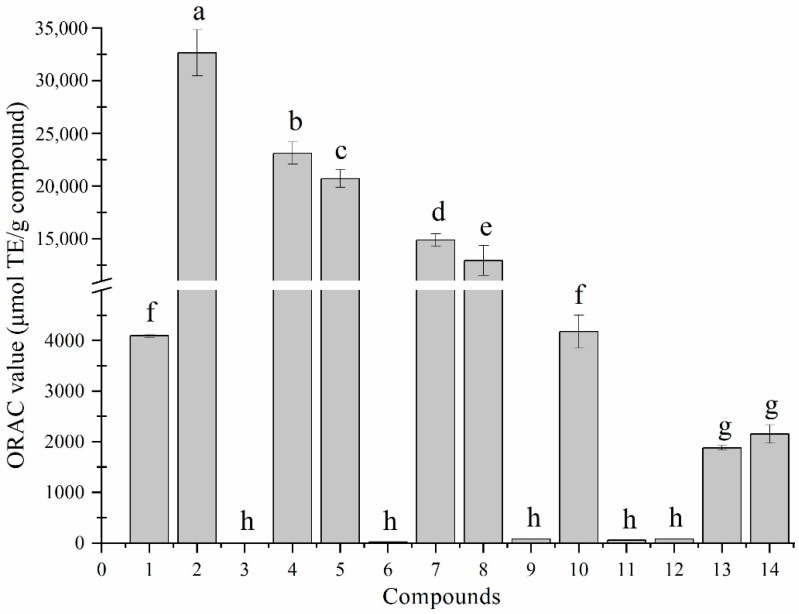
The ORAC values of compounds **1**–**14** (mean ± SD, n = 3). Bars marked with different letters mean significant difference at *p* < 0.05 between each other.

**Figure 3 foods-11-03131-f003:**
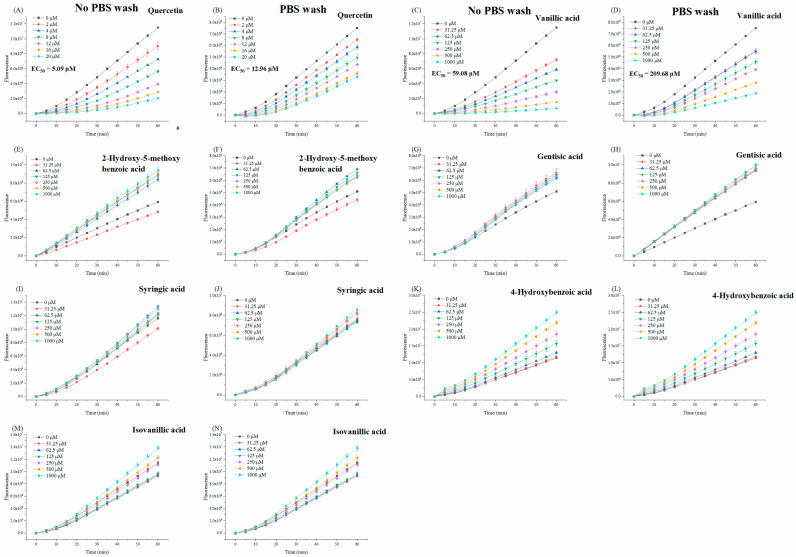
Cellular antioxidant activity of quercetin, vanillic acid, 2-hydroxy-5-methoxybenzoic acid, gentisic acid, syringic acid, 4-hydroxybenzoic acid and isovanillic acid without PBS wash (**A**,**C**,**E**,**G**,**I**,**K**,**M**) and with PBS wash (**B**,**D**,**F**,**H**,**J**,**L**,**N**).

**Table 1 foods-11-03131-t001:** The half maximal inhibitory concentration (IC_50_) of benzoic acid derivatives against α-amylase and α-glucosidase. Values (mean ± SD) with different letters within each column have a significant difference at *p* < 0.05.

No.	Compounds	IC_50_ (mM)	
		α-Amylase	α-Glucosidase
2	2-hydroxy-5-methoxybenzoic acid	40.50 ± 0.44 ^c^	26.08 ± 0.67 ^c^
4	gentisic acid	35.97 ± 2.23 ^b^	30.47 ± 1.02 ^d^
7	4-hydroxybenzoic acid	>100	34.07 ± 1.51 ^e^
8	vanillic acid	>100	29.72 ± 0.33 ^d^
13	syringic acid	>100	20.32 ± 0.39 ^b^
	acarbose	4.59 ± 2.01 ^a^	6.54 ± 0.91 ^a^

## Data Availability

The data presented in this study are available on request from the corresponding author.

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
