# Peer review of "Identification and Structure–Activity Relationship of Recovered Phenolics with Antioxidant and Antihyperglycemic Potential from Sugarcane Molasses Vinasse"

_foods, 2022, doi:10.3390/foods11193131_

Round 1

Reviewer 1 Report

Here I would like to send my comments about the manuscript "Identification and structure–activity relationship of recovered phenolics with antioxidant and antihyperglycemic potential from sugarcane molasses vinasse". The paper submitted in Foods was written with good quality. The methodology is objective and very well described. The results are interesting, and the discussion well-structured. The conclusion describes the work developed. The figures and tables are consistent with the work. I am favourable to accepting the article after minor revision.

1. The author should include a clear hypothesis in the introduction and a Evaluation of antioxidant activity in vivo.

Author Response

  1. The author should include a clear hypothesis in the introduction and a Evaluation of antioxidant activity in vivo.

Response: Thanks for your valuable suggestion. The potential antioxidant and hypoglycemic activities of phenolic acids were discussed in the introduction and discussion sections to emphasize the theme of the paper.

Reviewer 2 Report

Dear author your manuscript is presented in a good way and work indeed is new and as per current requirement for new waste disposal ad valorization however, you need to address  following comments:

1. Abstract required more data regarding significant findings and novelty of findings

2. Last paragraph of introduction must highlight your hypothesis

3. Materials and methods are excellent

4. results need to be concise

5. add latest reference where applicable

Author Response

  1. Abstract required more data regarding significant findings and novelty of findings.

Response: Thanks for your valuable suggestions and comments. The ORAC and CAA values of benzoic acid derivatives are shown in the abstract section to support our results.

  1. Last paragraph of introduction must highlight your hypothesis.

Response: Thanks for your valuable suggestions. Phenolic acids have potential antioxidant and hypoglycemic activities and are abundant in sugarcane molasses vinasse, which provides a clear significance for the recovery of phenolic acid from molasses vinasse waste. Therefore, we emphasize the importance of investigating the composition of phenolic acids in molasses lees and their potential antioxidant and hypoglycemic activities in the introduction section.

  1. Materials and methods are excellent.

Response: Thanks for your encouraging comments. In addition, we modified the materials and methods section to make it smoother.

  1. Results need to be concise.

Response: Thanks for your valuable suggestions. The discussion of ORAC activity and hypoglycemic activity is more concise. We remove unnecessary statements such as obvious ordering of values and data already in the table.

  1. Add latest reference where applicable.

Response: Thanks for this valuable suggestion. In the discussion section, we compare the ORAC values and α-amylase inhibition of benzoic acid derivatives with the results of recent reports (e.g., [28], [29] and [45] in the References).

Reviewer 3 Report

The manuscript presents interesting information on the recovery of bioactive compounds from sugarcane molasses vinasse. The manuscript needs revisions according to the suggestions listed below. 

The Introduction section is poor and needs to be completed with more information. 

L82: Please mention the year of acquisition. 

L124: Citation is missing (Wolfe).

L88: Please see the manuscript structure. 

L171-L316 There is no discussion of the results. Please include comparisons and explanations from the existing literature. 

L280: See axis name. 

Conclusion: Please include limitations and further perspective. 

Author Response

  1. The Introduction section is poor and needs to be completed with more information.

Response: Thank you for your valuable comments and suggestions. We highlight more points about the potential activity of phenolic acids in the Introduction section and further emphasize the hypotheses of this research.

  1. L82: Please mention the year of acquisition.

Response: Thank you for your reminders and suggestions and we have added information about the timing of the acquisition of raw materials in the Materials section.

  1. L124: Citation is missing (Wolfe).

Response: Thank you very much for your reminders. And the reference has been supplemented. We have added information about the references cited here and rechecked the formatting and citations of other references.

  1. L88: Please see the manuscript structure.

Response: Thank you for your suggestions. The missing subtitle number has been completed.

  1. L171-L316 There is no discussion of the results. Please include comparisons and explanations from the existing literature.

Response: Thank you for your valuable comments and suggestions. We compared the ORAC and α-amylase inhibition activity of benzoic acids with other recent results. Firstly, the ORAC activity of different extracts showed a high positive correlation with the content of hydroxybenzoic acids (vanillic, syringic, and gentisic acid), and our results further demonstrated the activity of these individual compounds. Secondly, it was further confirmed and explained that 2-OH on the benzene ring enhanced the α-amylase inhibitory ability of benzoic acid derivatives, and our work improved the structure-activity relationship regarding the hypoglycemic activity of benzoic acids. Unfortunately, since the CAA activity of benzoic acid derivatives has not been directly reported, we demonstrated a direct link between the CAA activity and the high vanillic acid content of antioxidant extracts.

  1. L280: See axis name.

Response: Thank you for your reminders. We revised the X-axis name to "Compounds" and corrected the font of the Y-axis name in Fig. 2.

  1. Conclusion: Please include limitations and further perspective.

Response: Thank you for these valuable comments. In the conclusion section, we proposed that the current study should evaluate the antioxidant and inhibitory activities of more benzoic acids with different structures (different numbers and sites of functional groups).

Round 2

Reviewer 3 Report

The authors tried to improve the manuscript.  However, the Introduction still needs revision by adding more information. Also, the results must be more discussed.

Author Response

The authors tried to improve the manuscript. However, the Introduction still needs revision by adding more information. Also, the results must be more discussed.

Response: Thanks again for suggestions on the content of our manuscript. We have listed all the changes and responses below:

In the Introduction section, a) we described the volume of molasses vinasse produced by ethanol production each year; b) we introduced the known main components of sugar cane molasses vinasse, including sugars, alcohols, organic acids, and amino acids; c) since organic acids and glycerol cannot be separated by distillation, we proposed the possibility of separating organic acids (including phenolic acids) from molasses vinasse; d) we illustrated the value of research on the recovery of valuable components from agricultural waste.

In the Discussion section, we additionally explained the key role and possible mechanism of 2-OH of benzoic acid derivatives in α-amylase inhibition.